# Nonlinear Changes in Botulinum Toxin Treatment of Task-Specific Dystonia during Long-Term Treatment

**DOI:** 10.3390/toxins13060371

**Published:** 2021-05-22

**Authors:** André Lee, Jabreel Al-Sarea, Eckart Altenmüller

**Affiliations:** 1Institute of Music Physiology and Musicians’ Medicine, University of Music, Drama and Media Hannover, Neues Haus 1, 30175 Hanover, Germany; eckart.altenmueller@hmtm-hannover.de; 2Department of Neurology, Klinikum Rechts der Isar, Technische Universität München, Ismaningerstr. 22, 81675 Munich, Germany; 3Hanover Medical School, Carl-Neuberg-Str. 1, 30625 Hanover, Germany; j.alsarea@yahoo.de

**Keywords:** botulinum toxin A, long term treatment, task-specific dystonia, nonlinear correlation, inter-injection interval, musician’s dystonia

## Abstract

Botulinum toxin (BoTX) is the standard treatment for task-specific dystonias (TSDs) such as musician’s dystonia (MD). Our aim was to assess the long-term changes in BoTX treatment in a highly homogeneous and, to our knowledge, largest group of MD patients with respect to the following parameters: (1) absolute and (2) relative BoTX dosage, (3) number of treated muscles, and (4) inter-injection interval. We retrospectively assessed a treatment period of 20 years in 233 patients, who had received a cumulative dose of 68,540 MU of BoTX in 1819 treatment sessions, performed by two neurologists. Nonlinear correlation was used to analyze changes in the parameters over the course of repeated treatments. Post-hoc we applied a median-split to classify two subgroups (high-BoTX, low-BoTX) depending on the total amount of BoTX needed during treatment. Across all patients, we found a decrease of dosage for the first approximately 25 treatments with an increase afterwards. The number of muscles and inter-injection intervals increased with time with a discrete decrease of inter-injection intervals after about 35 treatments. Subgroup differences were observed in the amount of BoTX and inter-injection intervals, with continuously increasing inter-injection intervals and decreasing BoTX dosage in the low-BTX group. Both groups showed a continuously increasing number of injected muscles. In summary, we found nonlinear changes of BoTX dosage and inter-injection intervals and a continuously increasing number of injected muscles with treatment duration in TSD-patients. Furthermore, we, for the first time, identified two subgroups with distinct differences. Increasing inter-injection intervals and decreasing BoTX dosages in the low-BoTX group indicated improvement of symptoms with continued treatment. Continually increasing BoTX dosages with unchanged inter-injection intervals in the high-BoTX group indicated deterioration.

## 1. Introduction

Musician’s dystonia (MD) is a focal task-specific movement disorder (TSD) originating in the central nervous system that occurs mainly in professional musicians [1,2,3] and leads to the loss of voluntary control of highly trained movements at the instrument. The most common form is an involuntary flexion of one or more fingers [4,5]. The pathophysiology is not yet sufficiently understood; however, recent findings suggest that it is a network disorder comprising the basal ganglia and cerebellum, as well as visuomotor and sensorimotor networks [6,7]. Lacking a causal therapy, the standard treatment for dystonias remains the injection of Botulinum toxin (BoTX) into the affected muscles [2,4,8,9].

The safety and efficacy of the BoTX treatment have been repeatedly reported [10,11,12,13,14,15,16,17,18,19]. Two recent papers assessed long-term safety and efficacy in two focal dystonias (Meige’s syndrome, cervical dystonia [18,19]); for both disorders, patients reported an increase in the duration of BoTX effects with repeated treatments. Another study looking at long-term effects of BoTX-treatment in blepharospasm, hemifacial spasms, and synkinesias after facial palsy found an increase of the *absolute* amount of BoTX for all disorders over the course of repeated treatments. This study also reported an increase in injection sites only for hemifacial spasm and synkinesias, but not for blepharospasm, and thus found a stable *relative* amount of BoTX (per injection site) for the first two disorders and an increase only for blepharospasm [16].

Another important parameter is the interval between two consecutive injections (inter-injection interval). Studies of cervical dystonia patients [20] and patients with spasticity [21] reported a preferred inter-injection interval of <10 weeks. However, the recommended inter-injection interval is 12 weeks [20,22]. This practice may lead to dissatisfaction if patients have to tolerate symptoms before the next injection [20,23]. In recent years, the practice of a flexible inter-injection interval has drawn attention [23,24]. At our outpatient clinic, rather than having injections every 12 weeks, we asked patients to make appointments for subsequent injections based on clinical demands, i.e., when they perceived increasing symptoms of dystonia and a loss of efficacy of the previous injection, irrespective of whether a 12-week interval was adhered to. Therefore, we were able to assess whether the inter-injection interval increased, decreased, or remained unchanged during long-term treatment.

Only a few studies have assessed long-term efficacy in TSDs, including one reporting on a variety of dystonias including 14 patients with writer’s cramp (WC) [10], and another study of 20 patients, but only 5 with MD [25]. The aim of this study was therefore to assess long-term patterns of BoTX treatment in a highly homogeneous group of patients with a TSD. Rather than assessing changes by comparing parameters at two points in time (e.g., at the beginning of follow-up and at the end of follow-up), we were interested whether continuous changes of four treatment parameters in the course of treatment exist. To our knowledge, continuous changes have not yet been assessed in the long-term treatment of dystonia. To this end, we assessed the relationships between treatment duration (i.e., number of treatments) and the (1) absolute and (2) relative amount of BoTX per treatment, (3) the number of muscles injected per treatment, and (4) the interval between consecutive treatments (inter-injection interval).

## 2. Results

### 2.1. Patients

(Mean age (±SD) of the 233 patients included was 51 ± 12 years. 82% were male.)

Applying the post-hoc median-split, we obtained the high-BoTX group (n = 124, age [mean ± SD] 52.9 ± 11.5 years) and the low-BoTX group (n = 109, age [mean ± SD] 47.9± 12.0 years). In the high-BoTX group we found a total of 54 treatments and in the low-BoTX group we found a total of 41 treatments (Table 1).

### 2.2. Absolute Amount of BoTX per Treatment:

#### 2.2.1. Absolute Amount of BoTX per Treatment for All Musicians

For all musicians, the median amount of BoTX was 26.7 MU (Q1–Q3: 18.0–40.0 MU) per treatment. There was a correlation between the amount of BoTX and treatment duration (r = 0.54; *p* = 0.020) (Figure 1, Table 2, for a boxplot with individual data see Appendix A).

#### 2.2.2. High-BoTX Group

In the musicians’ high-BoTX group, the median amount of BoTX was 69.2 MU (Q1–Q3: 54.2–96.7 MU) per treatment. There was a correlation between the amount of BoTX and treatment duration (r = 0.68; *p* = 1.69 × 10^−8^) (Figure 1, Table 2).

#### 2.2.3. Low-BoTX Group

In the musicians’ low-BoTX-group, the median amount of BoTX was 10.8 MU (Q1–Q3: 9.3–13.3 MU) per treatment. There was a correlation between the amount of BoTX and treatment duration (r = 0.68; *p* = 0.004) (Figure 1, Table 2).

### 2.3. Relative Amount of BoTX per Treatment

#### 2.3.1. All Musicians

For all musicians, the median relative amount of BoTX per treatment was 8.5 MU/muscle (Q1–Q3: 6.7–12.5 MU). There was a correlation between treatment duration and the relative amount of BoTX (r = 0.79; *p* = 2.50 × 10^−6^) (Figure 1, Table 2, for a boxplot with individual data see Appendix A).

#### 2.3.2. High-BoTX Group

In the musicians’ high-BoTX-group, the median relative amount of BoTX per treatment was 22.4 MU/muscle (Q1–Q3: 17.1–28.1 MU). There was a correlation between treatment duration and the relative amount of BoTX (r = 0.60; *p* = 1.37 × 10^−6^) (Figure 1, Table 2).

#### 2.3.3. Low-BoTX Group

In the musicians’ low-BoTX-group, the median relative amount of BoTX per treatment was 5.0 MU/muscle (Q1–Q3: 4.2–6.7 MU). There was a correlation between treatment duration and the relative amount of BoTX (r = 0.65; *p* = 5.17 × 10^−6^) (Figure 1, Table 2).

### 2.4. Number of Injected Muscles per Treatment

#### 2.4.1. All Musicians

For all musicians, the median number of injected muscles per treatment was 2.0 muscles (Q1–Q3: 2.0–3.6 muscles). There was a correlation between treatment duration and the number of injected muscles (r = 0.63; *p* = 2.71 × 10^−7^) (Figure 2, Table 2, for a boxplot with individual data see Appendix A).

#### 2.4.2. High-BoTX Group

In the musicians’ high-BoTX-group, the median number of injected muscles per treatment was 3.5 muscles (Q1–Q3: 3.0–4.0 muscles). There was a correlation between treatment duration and the number of injected muscles (r = 0.46; *p* = 4.71 × 10^−4^) (Figure 2, Table 2).

#### 2.4.3. Low-BoTX Group

In the musicians’ low-BoTX-group, the median number of injected muscles per treatment was 2 (Q1–Q3: 2.0–3.0 muscles). There was a correlation between treatment duration and the number of injected muscles (r = 0.49; *p* = 0.001) (Figure 2, Table 2).

### 2.5. Treatment Duration and Inter-Injection Interval

#### 2.5.1. Change of Treatment Interval for All Musicians

For all musicians, the median inter-injection interval was 4.1 months (Q1–Q3: 3.6–4.9 months). There was a correlation between the inter-injection interval and treatment duration (r = 0.27; *p* = 8.7 × 10^−4^) (Figure 2, Table 2, for a boxplot with individual data see Appendix A).

#### 2.5.2. High-BoTX Group

In the musicians’ high-BoTX group, the median inter-injection interval was 4.3 months (Q1–Q3: 3.5–4.7 months). There was no correlation between the inter-injection interval and treatment duration (r = 0; *p* = NA) (Figure 2, Table 2).

#### 2.5.3. Low-BoTX Group

In the musicians’ low-BoTX-group, the median inter-injection interval was 4.1 months (Q1–Q3: 3.6–4.9 months). There was a correlation between the inter-injection interval and treatment duration (r = 0.42; *p* = 0.007) (Figure 2, Table 2).

## 3. Discussion

In our study, we report on long-term BoTX treatment of a focal task-specific dystonia in 233 musicians who had received a cumulative dose of 68,540 MU of BoTX in 1819 treatment sessions. To our knowledge, this is the largest sample of long-term effects in a representative and highly homogeneous group of TSD patients reported for a treatment period of 20 years. Another strength of our study is the fact that injections were only given by two neurologists (EA, AL) and that almost none of the patients had been pretreated. In contrast, frequent changes of treating physicians and up to 34% of pretreated patients were found to be common for tertiary care hospitals [18,19]. Thus, our findings are likely to reflect changes caused by the course of dystonia since variance induced by changing physicians or a heterogeneous patient group was kept to a minimum. The focus of this study was to investigate continuous changes of four treatment parameters over the course of treatment rather than the safety and efficacy of BoTX treatment, which has been repeatedly reported [10,11,12,13,14,16,17,18]. To assess these changes, we applied a recently introduced nonlinear correlation method, which allowed us to look at correlations (i.e., continuous changes over the entire treatment period) rather than comparing the first and last treatment [10,25,26,27,28] or different points in time during the treatment as in previous studies [16,29].

One limitation of the study is that we did not include the patients’ functional stage with regard to the disorder, and thus can only speculate about the relation between the amount of BoTX or time interval and the severity of the disorder. Further prospective studies are needed that include clinical parameters such as severity of symptoms or age at onset of dystonia.

### 3.1. Amount of BoTX per Treatment and Number of Muscles Injected per Treatment

In comparison with previous reports of long-term BoTX treatment in focal dystonias, the median amount of BoTX in our sample (26.7 MU) was slightly lower than in blepharospasm [10,16,17,19,29], and markedly lower than in cervical dystonia [10,11,14,18]. It was also lower than in one long-term study on 14 patients with WC (30–200 MU) [10] and in earlier efficacy studies on WC [30,31], and comparable to only one long-term study of TSD including five musicians (25–50 MU) [25]. However, follow-up duration in both long-term studies was only 10 years and no specific BoTX dosage in musicians in the latter study was reported. Furthermore, our median dosage was higher than the recommended dosage of 19.9 MU BoTX for the finger flexor muscles in WC in the recently published consensus guideline for BoTX therapy [32].

We found a nonlinear change in the amount of BoTX, with a decreasing amount for approximately the first 25 treatments and a subsequent increase. This is a surprising finding as most other studies reported an increasing amount of BoTX [10,16,25,26,27,28,29]. However, two recent studies on cervical dystonia and blepharospasm/Meige’s syndrome found increasing BoTX dosages in later stages of treatment, albeit after several years of stable amounts without any decrease at any moment during treatment [18,19].

In order to investigate possible reasons for the nonlinear distribution, we applied a post-hoc median-split and compared the two subgroups, i.e., high- and low-BoTX groups. This approach yielded several interesting findings: Firstly, we found a longer treatment duration for the high BoTX-group (54 treatments vs. 41 treatments). Secondly, we found that in the high-BoTX group, continuously increasing amounts of BoTX were necessary to yield the desired effect, whereas in the low-BoTX group, a decrease was found until the 30th treatment, after which an increase was seen. One possible explanation for increasing absolute amounts of BoTX might be increasing the number of muscles treated with BoTX. Likewise, decreasing amounts of BoTX might be caused by a decreasing number of treated muscles. Interestingly, one recent study that reported an increase of BoTX amounts in long-term treatment in three disorders (blepharospasm, facial hemispasm, and facial synkinesias) found that while the absolute amount of BoTX increased in all three disorders, the relative amount per injection site increased only in blepharospasm but was unchanged in facial hemispasms and facial synkinesias, due to an increase in injection sites only in the latter two disorders [16]. We therefore looked at the number of injected muscles during the course of treatment for all patients and in the two subgroups and found a continuously increasing number of injected muscles for all patients as well as in both subgroups. This indicated that dystonic symptoms spread to other fingers with longer disease duration. Thus, the increase in the absolute amount of BoTX needed to obtain a desired effect in the high-BoTX group may be at least partly explained by the spreading of dystonia to other fingers. However, since the (relative) amount of BoTX per muscle continuously increased as well, each affected muscle necessitated increasing amounts of BoTX in a superadditive way. One feasible assumption is that either the severity of dystonic symptoms increased in addition to the spreading of dystonia, or that a tolerance for BoTX led to a reduced efficacy, thereby necessitating higher doses [25,27] (or a combination of both). Why this affected only the high-BoTX group remains unexplained. The development of antibodies is unlikely, since this should have resulted in a complete treatment failure [33], which we did not observe.

In contrast, the low-BoTX group needed continuously lower absolute amounts of BoTX despite the fact that in the course of treatment more fingers were affected. Consequently, the (relative) amount of BoTX per muscle continually decreased in that group. One possible explanation is that musicians in the low-BoTX group had a less severe dystonia and/or a good response to BoTX, resulting in a continuing improvement of playing ability until BoTX was no longer necessary. This assumption is corroborated by the fact that in the low-BoTX group, patients had fewer treatments than in the high-BoTX group (41 vs. 54 treatments, respectively). Given our findings, assessing the relative amount of BoTX per muscle (or injection site) yields important additional information on the course of the disease [16], and it cannot be ruled out that the increase in BoTX dosage reported in the above-mentioned studies [10,25,26,27,28,29] may be due to an increase in the number of injected muscles. In comparison to other focal dystonias, the relative amount of BoTX we administered to all patients (8.5 MU) was lower than in cervical dystonia (approx. 50 MU) [34] but higher than in hemifacial spasm, facial synkinesias, and blepharospasm (2–7 MU) reported by Laskawi et al. 2014 [16].

Future prospective studies need to address the role of symptom severity on the amount of BoTX in order to allow for an early classification of the patients into either group, since this would have an important impact on the prognosis of dystonia, the counseling of patients, and in the therapeutic planning.

### 3.2. Inter-Injection Interval

The median inter-injection interval we found (4.1 months) was much shorter than the mean interval of 19.9 months reported for five musicians [25], and comparable to inter-injection intervals for focal dystonias and facial hemispasms of 21 weeks in one long-term study (≈4.8 months) [29]. It is, however, longer than the effect duration reported in WC (2.4–3.2 months) [35] and one study in cervical dystonia (14.8 weeks ≈ 3.4 months) [36].

We found a nonlinear change of inter-injection intervals with an increase for all patients for approximately the first 35 treatments and a slight decrease afterwards. Reports of inter-injection intervals or effect duration are inconsistent in prior studies. For blepharospasm, increasing inter-injection intervals [28,29] as well as unchanged effect duration has been reported [19]; in cervical dystonia, decreasing inter-injection intervals [29] or unchanged effect duration of 10–11 weeks was found [18]. Yet another long-term study assessing different focal dystonias found increasing effect durations [27].

Comparison of the two subgroups again revealed distinct differences: in the high-BoTX-group, we found unchanged inter-injection intervals in the course of treatment, whereas in the low-BoTX-group, continuously longer inter-injection intervals were seen. The reason for this finding remains speculative and we offer the following explanation: longer intervals between treatments in the low-BoTX group may reflect an improvement in playing ability leading to longer periods of sufficient playing ability. This is corroborated by the above-described finding of decreasing amounts of BoTX in the low-BoTX group, which may reflect an improvement. Vice versa, a more severe dystonia in the high-BoTX group with an increasing number of affected muscles and increasing amounts of BoTX, one would not expect increasing inter-injection intervals.

More flexible inter-injection intervals have been discussed lately [23] and have been found to be safe and effective [14,24]. Those studies discussed inter-injection intervals of less than 3 months, since recurring symptoms before 12 weeks or even a return to pre-injection severity of symptoms was reported by patients [20,36], and a restriction of inter-injection intervals was found to be “an impediment to both patient satisfaction and treatment outcomes” ([21] as cited in [23]). In our patients, shorter inter-injection intervals while dystonic symptoms are still well-controlled could increase the risk of an undesirable weakness. Our finding of a nonlinear course of the inter-injection intervals supports the idea of flexible inter-injection intervals that adapt to the needs of patients and is based on a shared decision of patients and physicians by including clinical criteria (e.g., recurrence of symptoms as perceived by the patient and as estimated by the clinician or side effects). Again, prospective studies are needed to identify predictors for the inter-injection interval during treatment with BoTX.

## 4. Conclusions

We demonstrate that BoTX dosage and inter-injection interval change in a nonlinear fashion during long-term treatment of a TSD. Our findings support the concept of flexible, patient-oriented inter-injection intervals. The number of injected muscles continuously increased with treatment duration, indicating that dystonia spreads to other muscles in the course of the disease.

We also demonstrate that the nonlinearity can partly be explained by the identification of two subgroups with distinct differences. The low-BoTX group mainly required continuously lower BoTX dosages with increasing inter-injection intervals, likely reflecting an improvement of dystonia. In contrast, the high-BoTX group needed continually increasing amounts of BoTX to an increasing number of muscles with unchanged inter-injection intervals, likely reflecting an unfavorable course of dystonia.

Future prospective studies should investigate clinical parameters that can serve as predictors and allow for a classification of each MD patient early in treatment into either group. This knowledge would have a significant impact on the counseling of patients with regard to their prognosis and treatment plan.

## 5. Material and Methods

This study was a retrospective data analysis of patients with MD of the hand/fingers, treated with injection of BoTX in our outpatient clinic between 1994 and 2014. Injection was sonographically or electromyographically guided. We diluted 100 mouse units (MU) of either Onabotulinumtoxin (ONA, Botox^®^, Allergan GmbH, Frankfurt/Main, Germany) or Incobotulinumtoxin (INC, Xeomin^®^, Merz Pharma GmbH & Co. KGaA, Frankfurt/Main, Germany) or 500 MU Abobotulinumtoxin (ABO, Dysport^®^, Ipsen Pharma GmbH, Ettlingen, Germany) in 1 mL sodium chloride solution. We usually had one injection site per muscle. In subsequent appointments made by the patients based on clinical demands we discussed desired effects and side effects and the dosage of BoTX was adjusted. Likewise, if the number of fingers affected by dystonia changed, we adjusted the number of injected muscles accordingly.

### 5.1. Patients

We included only files of patients who had received at least two injections of BoTX. We obtained data about the musical instrument, the affected side, and the direction of dystonia (flexion or extension). Musicians with embouchure dystonia were excluded. We included 233 patients with MD of the upper extremity who had received BoTX at our outpatient clinic at least twice. Patients’ details are given in Table 3.

### 5.2. Data Acquisition and Analysis

For each patient, we extracted from the patient file the date of each treatment, thereby obtaining the inter-injection intervals in months, the total amount of BoTX given in MU per treatment, the number of injected muscles at each treatment, and the product given (ONA, Allergan GmbH, Frankfurt/Main, Germany; ABO, Ipsen Pharma GmbH, Ettlingen, Germany; or INC, Merz Pharma GmbH & Co. KGaA, Frankfurt/Main, Germany). MU ABO were converted to comparable MU of ONA or INC with the ratio 3:1 [37,38]. We calculated the median amount of BoTX for each treatment, the median number of injected muscles, the median relative amount of injected BoTX (per muscle)*,* and the median inter-injection interval.

Post-hoc we assessed two subgroups of patients by applying a median-split: a high-BoTX group that needed above-average amounts of BoTX and a low-BoTX group that needed below average amounts of BoTX.

### 5.3. Statistics

Because of the nonlinear distribution of our data, we applied a recently developed nonlinear correlation test. This test finds nonlinear correlation between two variables that may usually be overlooked by linear correlation tests by using an adaptive local linear correlations computation. It returns a nonlinear correlation estimate ranging from 0 to 1, with higher values indicating a higher correlation; negative correlations are not applicable. The *p*-value is adjusted to avoid false positive results [39,40]. We used R version 3.1.3. A correlation was assessed between the treatment duration (i.e., number of treatments) and (1) the amount of BoTX per treatment, (2) the number of muscles injected per treatment, and (3) the interval between consecutive treatments. The significance level was set at α = 0.05.

## Figures and Tables

**Figure 1 toxins-13-00371-f001:**
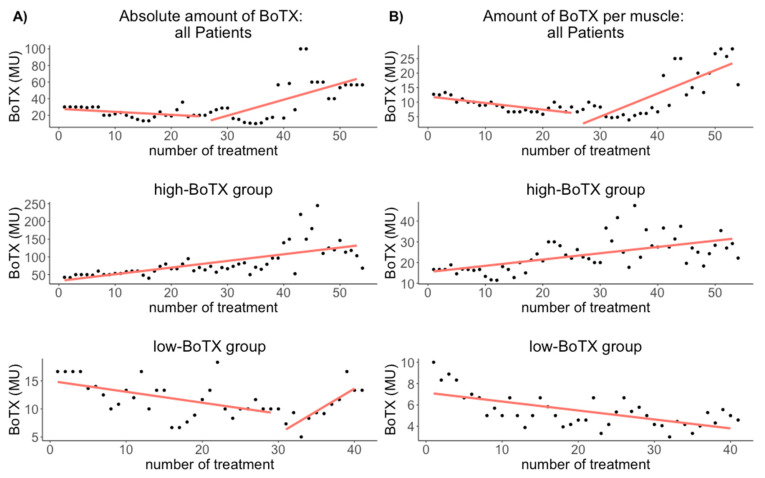
Results of the nonlinear correlation estimation between the treatment duration and the median absolute amount of BoTX (**A**) and the median relative amount of BoTX (**B**) for all patients (upper row), the high-BoTX group (middle row), and the low-BoTX group (lower row). In the high-BoTX group there is a constant increase in the absolute and relative amount of BoTX ((**A**,**B**) middle row). In the low-BoTX group there is a decrease until the 30th treatment with a subsequent increase for the absolute amount of BoTX ((**A**) lower row), and a continuous decrease in the relative amount of BoTX ((**B**) lower row). In the high-BoTX group there are 54 treatments and in the low-BoTX group there are 41 treatments. The absolute and relative amount of BoTX for all patients decreases approximately until the 25th treatment, after which it increases again ((**A**,**B**) upper row). One reason for this increase is likely the longer treatment duration in the high-BoTX group, in addition to the increasing absolute amount of BoTX after the 25th treatment in that group ((**A**) lower row). BoTX: Botulinum toxin A = MU = Mouse Units.

**Figure 2 toxins-13-00371-f002:**
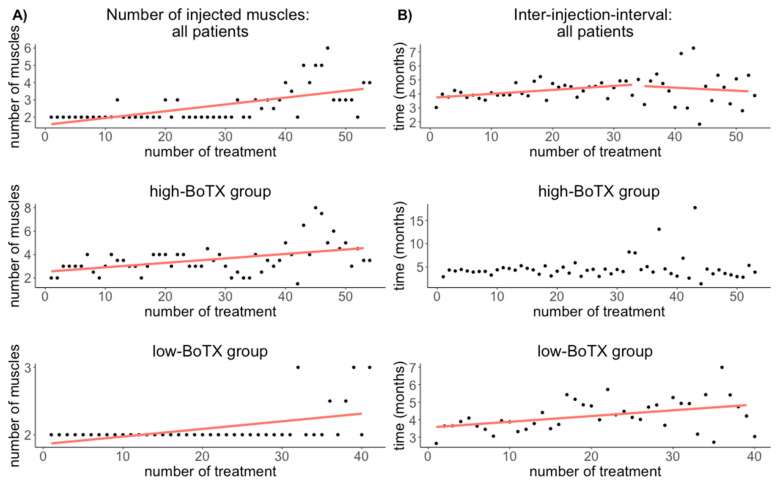
Results of the nonlinear correlation between the treatment duration and the median number of injected muscles (**A**) and the inter-injection interval (**B**) for all patients (upper row), the high-BoTX group (middle row), and the low-BoTX group (lower row). In the high-BoTX group no correlation was found between treatment duration and the inter-injection intervals ((**B**) middle row) and in the low-BoTX group the inter-injection intervals increased constantly ((**B**) lower row). In the high-BoTX group, there are 54 treatments and in the low-BoTX group, there are 41 treatments.

**Table 1 toxins-13-00371-t001:** Demographic data for the sub-groups (high- and low-BoTX): mean and SD for age; absolute and relative number for gender. SD = standard deviation.

	High-BoTX Group		Low-BoTX Group	
	Mean	SD	Mean	SD
Age (years)	53	11.5	48	12.0
	n	%	n	%
Gender				
Female	14	11.3%	26	23.85%
Male	110	88.7%	83	76.15%

**Table 2 toxins-13-00371-t002:** Results of the nonlinear correlation with the estimated correlation coefficient r and the adjusted *p*-value; median and interquartile range of the four treatment parameters for all patients and the subgroups. BoTX: Botulinum toxin; High-BoTX: Patients in the high-BoTX group; Low-BoTX: Patients in the low-BoTX group; NA: not applicable; Q1–Q3: 1st quartile—3rd quartile.

	r	*p*	Median	Q1–Q3
Absolute amount of BoTX				
All patients	0.54	0.020	26.7	18.0–40.0
High-BoTX	0.68	1.69 × 10^−8^	69.2	54.2–96.7
Low-BoTX	0.68	0.004	10.8	9.3–13.3
Relative amount of BoTX				
All patients	0.79	2.50 × 10^−6^	8.5	6.7–12.5
High-BoTX	0.60	1.37 × 10^−6^	22.4	17.1–28.1
Low-BoTX	0.65	5.17 × 10^−6^	5.0	4.2–6.7
Muscle				
All patients	0.63	2.71 × 10^−7^	2.0	2.0–3.6
High-BoTX	0.46	4.71 × 10^−4^	3.5	3.0–4.0
Low-BoTX	0.49	0.001	2.0	2.0–3.0
Inter-injection interval				
All patients	0.27	8.7 × 10^−4^	4.1	3.6–4.9
High-BoTX	0	NA	4.3	3.5–4.7
Low-BoTX	0.42	0.007	4.1	3.6–4.9

**Table 3 toxins-13-00371-t003:** Demographic and music-related data: mean and SD for patients’ age; absolute and relative number for gender; side affected by dystonia; direction of dystonia; and instrumental group.

	Mean	SD
Age (years)	50	12
	n	%
Gender		
Female	40	17%
Male	193	83%
Side affected		
right	142	61%
left	91	39%
Dystonia direction		
flexion	206	88.4%
extension	24	10.3%
both	3	1.3%
Instrumental group		
Keyboard	83	35.6%
Woodwind	56	24.0%
Plucked	50	21.5%
String	35	15.0%
Percussion	6	2.6%
Brass	2	0.9%
Conductor	1	0.4%

## Data Availability

The data presented in this study are available on request from the corresponding author. The data are not publicly available due to data protection.

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
