# Peer review of "Nonlinear Changes in Botulinum Toxin Treatment of Task-Specific Dystonia during Long-Term Treatment"

_toxins, 2021, doi:10.3390/toxins13060371_

Round 1

Reviewer 1 Report

The manuscript titled "Long-term continuous effects of botulinum toxin treatment of activity-specific dystonia" needs a thorough review by a native English. I hope that this revision will greatly ameliorate the reading and help the authors to improve the text which, at present, consists of numerous repetitions of the data obtained by the authors themselves and those present in the literature, while the experimental and clinical importance of the data obtained by the authors  is not adequately highlighted and discussed

Abstract

please, avoid to write  as a single word botulinum toxin. I suggest to use the common acronym: BoTX all along the text. Indeed, a few rows below  the authors write BTX. Thus, there is confusion. Please make order and choose the common acronym I suggested

Write separately:  ‘standard treatment’ in the entire text

Row 6: write: following parameters: (avoid to write twice: treatment)

Row 7: avoid the acronym iii for inter-injection-interval. Write: inter injection interval always

Row 8: write ‘..1819 injections..’

In the whole, the abstract is not explicative

  1. Introduction

From row 50 to row 54: it is not clear what the authors are talking about. Especially the last sentence does not make any sense to me. Please make the sentences comprehensible.

Row 62:’…and which of our knowledge has not been done so far.’ What do the authors mean?

  1. Results

2.1 Patients

From line 68 to line 77 the content belongs to the methods. Furthermore, all the information is present in Table 1 which, again, belongs to the chapter on methods. Thus, the discursive part could be shortened

  1. Discussion

Row 174 Why the authors change the definition of the disease? (see Abstract and Introduction). Please, use the same definition always.

Rows 216-217: I do not understand what the authors mean with ‘..(or viceversa).’

  1. Conclusions

This chapter is mainly a repetition of the results obtained.

Reviewer 2 Report

This was interesting study. However, there were numerous typo errors and improper usage of words. They could be corrected by professional proof-reading service. I suggested some examples of poor English. The main finding of this paper was that low BTX group showed less frequent treatment than high BTX group. When looked at Figure 2, low BTX group received injection to the smaller number of muscles. If low BTX group received longer time of treatment, the number of treatment was also increased. Therefore, the main findings seemed to be associated with the severity of the disease. As mild diseased patients received low dosage of BTX, they might require less frequent treatment. Therefore, the data for the disease severity should be included for the analysis.

Abstract:

Page 1, Line 4, 6, 8, 11, & 16. There should be a space between "Botulinum" and "toxin".

Page 1, Line 12, "The number of muscles and iii increased" What's "iii"? If it was inter-injection interval, please use capital, "III". It was confused to figure, 3.

There was unclear logic flow between the results and the conclusion. What's "early classification"? How the patient classification is associated with BTX dosage?

Key Contribution section:

What's "knowledge largest group"?

What's the meaning of this sentence, "an initial decrease of Botulinumtoxin dosage with a subsequent increase"? Is it "initial low dosage of BTX and subsequent increase of BTX dosage"?

What's "mainly continuously increasing inter-injection-intervals"? Is it meant, "general tendency of widening inter-injection-intervals"?

What's "amounts of muscles"? Is it muscle volume or number of muscle fiber?

Main text

Table had typo errors.

Round 2

Reviewer 1 Report

I carefully read the revised version of the manuscript titled: ‘Nonlinear changes in Botulinum toxin treatment of task-2 specific dystonia during long-term treatment’ and I found it significantly improved in all parts.

Reviewer 2 Report

This manuscript has been revised successfully.